# Performance Improvement of Construction Workers to Achieve Better Productivity for Labour-Intensive Works

Emmanuel Bamfo-Agyei [1,2,*], Didibhuku Wellington Thwala [2] and Clinton Aigbavboa [3]

1. Department of Construction Technology and Management, Cape Coast Technical University, Cape Coast P.O. Box DL 50, Ghana
2. Department of Civil Engineering, College of Science, Engineering and Technology, University of South Africa, UNISA, 0003, Pretoria P.O. Box 392, South Africa
3. Department of Construction Management and Quantity Surveying, University of Johannesburg, Johannesburg P.O. Box 524, South Africa
* Correspondence: emmanuel.bamfo-agyei@cctu.edu.gh

**Abstract:** This paper develops techniques to improve labour productivity in the construction industry and determine the level of labour productivity in the Ghanaian construction industry. The goal of this study was to develop a framework for determining the optimal productivity of construction workers for labour-intensive projects. There were three main objectives of this study: to identify factors that influence construction labour productivity in Ghana, to determine techniques used to improve construction labour productivity, and to develop a comprehensive framework for improving construction labour productivity in Ghana. The study adopted a quantitative research design that used a questionnaire. Since the country has been divided into zones, a stratified sampling technique was used based on the diverse nature of the population. Meanwhile, since the district offices were not all handling road construction projects, a purposive sampling technique was used to select 40 districts that were involved in road construction projects. A total of 560 respondents were sampled for the study. The data obtained from the study were analysed and are presented in tables and diagrams. The following factors played a significant role: the age of beneficiaries, the knowledge of beneficiaries, compliance with safety regulations, and the motivation of beneficiaries. Growing a project's beneficiary base has been observed by many sites to be associated with a decrease in overall labour productivity (due to the overcrowding of workers). Recruiting new members should be conducted cautiously, as the government plans to use this medium to benefit the impoverished in the region. Construction workers can use this information to aid in firm decision-making. For planning purposes, this research can also be used as a useful tool for utilizing labour-intensive methods to increase productivity and meet contract deadlines by finishing a task as anticipated.

**Keywords:** construction industry; Ghana; labour intensive; productivity; road

## 1. Introduction

Productivity loss is a major issue in the construction industry in developing countries because of the absence of documented data for project estimation, planning, and management. Constant concerns have been expressed about the lack of precise statistics on the industry and the labour productivity of its sub-sectors [1]. Considering that many construction projects require a significant quantity of labour, the issue of worker productivity becomes particularly important because higher productivity levels often equate to greater profitability, competitiveness, and earnings [2]. Work teams connected with various trades, levels of schooling, and weather conditions are all involved in the construction of a project [3].

The author of [4] argued that the use of locally accessible input (local labour) in labour-intensive programmes creates more demand for local products and services than does the use of imported technology and equipment in high-technology programmes. The

industry's success and growth are hampered by a lack of meaningful quantitative data. Construction labour productivity can be affected by geographic location, according to [5]. For decades, contractors have had to deal with the difficulty of fluctuating labour output rates in the construction sector, which has resulted in inaccurate contract period estimates due to inaccurate activity durations. According to [6], the primary cause of low labour outputs is poor management, and the authors found that a lack of alignment among goals and a lack of attention on the labour force are among the obstacles to the enhancement of intense labour outputs.

Understanding how various factors affect labour productivity is essential, since it is more variable and unpredictable than other project cost components [7]. The cost of labour can be reduced in direct proportion to an increase in productivity. As far as Ghanaian construction labour productivity frameworks are concerned, there has been very little research conducted in the area of understanding the knowledge of beneficiaries involved in labour-intensive projects. The goal of this study was to develop a framework for determining the optimal productivity of construction workers for labour-intensive projects. There were three main objectives of this study: to identify worker component factors that influence construction labour productivity in Ghana, to determine techniques used to improve construction labour productivity, and to develop a comprehensive worker component framework for improving construction labour productivity in Ghana.

## 2. Theoretical and Conceptual Frameworks of Labour Productivity

Construction sector labour productivity is examined from a variety of theoretical and conceptual viewpoints in this section. Theories of labour productivity that have been around for a long time, as well as more contemporary ones, were examined. Both the industry and activity levels were examined in terms of factors influencing labour productivity. How different circumstances impact the efficiency of construction workers was also taken into account.

In the context of infrastructure projects, the term "labour-intensive strategy" refers to the use of labour as the primary resource while guaranteeing cost-effectiveness and preserving quality. Taking a "labour-intensive strategy" means making the best use possible of human labour as a primary resource in infrastructure construction while also keeping an eye on both cost and quality.

This necessitates a well-balanced mix of manual labour and lightweight equipment [4], which includes ensuring that labour-intensive projects do not turn into "make-work" projects, in which both the costs and quality are overlooked. Labour-intensive works create far more job opportunities per unit of spending than does capital-intensive construction. The use of labour-intensive methods during infrastructure construction and maintenance is something that should be encouraged to help alleviate poverty and create jobs.

### 2.1. Theoretical Framework

The theoretical framework for this study was based on the labour productivity frameworks developed in [8,9]. The latent variables of the theory described in [8] are the workers and the materials as the major components for determining construction labour productivity. Similar findings have been made by the authors of [9], who found that labour productivity is influenced by the appropriateness of the materials and worker expertise. Both frameworks have a solid theoretical foundation and have been used to conceptualize a wide range of construction labour productivity within a broader theoretical framework, which is why they were an excellent starting point for this study.

This study's conceptual framework was heavily influenced by the approach taken in [9], which viewed construction labour productivity as both a criterion for assessing the productivity of businesses and as a criterion variable that predicts the productivity of firms. Thus, construction labour productivity was considered a "criterion variable," and therefore a dependent variable. This method, which was also employed in [10,11], was applied in the current investigation.

All previous theoretical foundations were built on the fundamental factors and constructs associated with the current conceptual framework. However, this research took into account the impact of temperature and an awareness of work-based circumstance components on construction labour productivity. These are the exogenous variables that play a role in determining the overall labour productivity, which is the endogenous variable.

For this framework to be effective, it must be applied to the Ghanaian building industry. Whether or not the industry is dependent on the stated features of the variables must be determined by evaluating the impacts of temperature and comprehending the effects of work-based conditions on labour productivity in the construction business. The theoretical framework proposes that the interaction between exogenous variables, such as the fundamental elements by which the subjective and objective measurements are related, is what determines the productivity of construction labour. Labour productivity in construction is heavily influenced by the variables examined in this review. To accommodate the unique features of the Ghanaian building industry, these have been altered somewhat. It is only after taking into account both the objective and subjective data that the previously discussed definition of productivity in the Ghanaian construction industry can be fully understood.

The theoretical underpinnings of this priority can be found in [8,9]. Construction labour productivity frameworks and approaches were implemented, such as those used in [10–15].

Accordingly, while the primary variable under examination is firm productivity and its relationship to other external variables, discussing it without these considerations is nearly impossible. Ghana's subjective assessment of its productivity, as defined by the construction sector, is one way in which labour productivity is conveyed. Finding the most important elements depends on the data available, and that data may vary depending on the situation. For example, sector-specific factors are thought to influence how organizations judge productivity in a certain industry. To be inclusive of all the experiences of the aspects of the Ghanaian construction industry that influence their evaluations, this document has been developed. The objective evaluation of construction labour productivity in this study was measured by assessing the actual performance of enterprises on labour productivity results.

*2.2. Factors Affecting Labour Productivity on Projects*

Numerous studies have been conducted on the topic of construction labour productivity. According to [16], numerous variables can affect a project's labour productivity. There are a total of 13 factors that affect productivity, according to [17]. Site layout, construction information complexity, the percentage of work performed by subcontractors, and supervisory quality were all included in this collection of 13 criteria, which is known as the management component.

When it comes to the workers, it is important to consider their level of training and experience as well as their overall size, composition, and the number of tasks they are responsible for performing while on the job. An important factor in creating favourable working conditions is determining how employees go about their daily tasks, as well as how long each day lasts.

Five factors have been recognized as affecting labour productivity: management (inspection delay), equipment and tools (lack of adequate tools and equipment), workers (work safety), and external factors (site circumstances and lack of materials). As stated by the authors of [18], the most significant risk to the contractors is the possibility of losing output due to a lack of adequate supplies, equipment, and manpower.

Construction productivity in Thailand is negatively affected by some factors, including a lack of materials and a lack of competent supervisors, as well as a lack of complete drawings, long instruction periods, a disorganized site layout, and long inspection times, as identified in [19,20]. When it came to determining which components were the most crucial, it came down to a lack of tools and equipment, worker absenteeism, and work-based conditions for rework. According to [21], the following factors have been found to have

the greatest impact on labour productivity: a shortage of materials and delays in material arrivals, which were highlighted as two problems in this category.

In terms of management, there were a variety of issues, such as ambiguous instructions given to workers, design changes, labour strikes, financial difficulties, a lack of oversight, and supervisors who were more likely to be absent than the workers themselves. Productivity was shown to be influenced by the availability and quality of tools and equipment. An investigation of the factors that affect the productivity of construction workers was conducted by [22]. There is a strong consensus in Nigeria that management and control (inadequate payment of completed works, and inadequate experience and managerial training) and materials (shortage of materials due to fluctuations, long delays, and delivery uncertainties, as well as inadequate logistics) are significant factors influencing productivity.

Worker supervision, the absence of construction manager leadership, and the extent of variations/change orders during the execution were recognized as the three management components that had the greatest impact on labour productivity by the authors of [23], who focused on the relative importance index. The technique of construction was incorporated in the work-based condition component. Delays in payment, worker tiredness, lateness, early departure, and frequent unscheduled breaks, as well as worker skill and the availability of experienced workers, were all recognized as factors affecting the productivity of the workforce. According to [24], five main components influence the fluctuation of productivity in the workforce. Employees, management, materials, equipment, and tools, as well as work-based conditions, were all covered.

The findings related to labour productivity at the construction level are largely based on studies in developed countries; very little is known about labour productivity in developing countries. This is evident by using the task frameworks developed in [8] and the simulation frameworks developed in [9]. In addition, compared to studies conducted in rich nations, the few studies that concentrated on emerging countries have not effectively offered an overview of the idea of labour productivity.

With regard to construction labour productivity in developing countries, there will be inconsistencies in the findings and the application of frameworks designed for developed countries that are applied in developing countries [25].

Temperatures in outdoor working situations can be more dangerous for workers engaged in labour-intensive public works on rural road construction projects. The output of the workers may suffer if they are forced to labour outside in temperatures above 32 degrees Celsius, as their bodies immediately reduce their activity to avoid overheating. The weather has a degree of unpredictability to it. A lack of planning might lead to weather-related delays and damage that necessitates rework. There are a variety of factors at play when working in less-than-ideal weather conditions.

This study is based on the assumption that productivity could not be achieved without the influence of understanding the attitude of beneficiaries to work components on labour productivity in Ghana. This is because productivity in a given industry is not determined by only one set of factors, but rather is influenced by a variety of factors.

## 3. Materials and Methods

### 3.1. Research Design

The study adopted the positivism approach in developing a labour productivity framework for the labour-intensive work of feeder road construction in Ghana. The use of statistical analysis, measures of association, and the development of measurement models are significant in this approach.

Hence, in this study, researchers employed a questionnaire-based descriptive survey to gather information about factors that affect the productivity of labour-intensive tasks [26]. This strategy is beneficial for researchers, since it allows them to generalize their findings from a specific group of people [27–29].

Principal component analysis (PCA) was used to analyse and decrease the observed variables to smaller elements that are crucial for labour productivity. For the purpose

of distilling the data into a manageable number of components, the researchers of [30] asserted that PCA can be used to extract factors based on the highest eigenvalues.

### 3.2. Study Population

The target population for the study was all contractors, site engineers, facilitators, timekeepers, district engineers, and Ghana Social Opportunity Project (GSOP) desk officers. Records available at the GSOP indicated that there were 920 professionals involved in labour-intensive works. These comprised 200 contractors, 200 site engineers, 200 facilitators, 200 timekeepers, 60 district engineers, and 60 GSOP desk officers.

### 3.3. Sampling Techniques and Sample Size

Bolgatanga, Wa, Temale, Kumasi, and Accra are the five regional Ghana Social Opportunity Project hubs. As a result of the demographic heterogeneity of the country's population, a stratified sampling method was employed to collect survey responses from each of the zones. Bolgatanga has 12 district offices, Wa has 10, Temale has 11, Kumasi has 14, Accra has 13, and there are a total of 60 district offices serving the various zonal offices.

Purposive sampling was utilised to choose 40 districts that were actively participating in road construction projects, because not all district offices were responsible for such work. A total of 120 sites were randomly selected to participate. In total, 560 respondents participated in the survey, and the authors of [31] acknowledge that this is a good sample size.

### 3.4. Research Instrument

The questionnaire was the primary method of collecting data [32]. Some items on labour productivity used in the questionnaire were extracted from reviews of the literature, and others were developed by the researchers, resulting in the compilation of a questionnaire divided into four sections.

### 3.5. Data Collection

From November 2016 through August 2017, 560 questionnaires were sent to potential respondents who perform labour-intensive tasks on road-building projects in Ghana using the drop-and-collect approach.

The questionnaire was in three sections (i.e., A, B, and C). The first section obtained demographic information about the respondents' personal information such as age, gender, occupation, educational background, the experience level of respondents, and their geographical location.

Section B sought to identify factors that influenced their company's productivity. The questionnaire items in all sections outside Section A used a five-point Likert scale. We used a five-point scale ranging from "excellent" (E) to "very poor" (VP) to denote the quality of each submission.

Company productivity can be affected by a number of different factors, and it can be measured in a variety of ways. Respondents were asked to indicate how strongly they agreed or disagreed with each statement about these factors and measures by marking the corresponding checkbox (X) or writing in the appropriate response (where E = 5, G = 4, A = 3, P = 2, and VP = 1) [33,34].

### 3.6. Data Analysis

Descriptive data analyses and multivariate correlational data analyses, including exploratory factor analyses, were conducted. The data obtained from the study were analysed and are presented in tables and diagrams. The Kaiser–Meyer–Olkin (KMO) test [35–39] and Bartlett's test of sphericity [40] were conducted to determine the suitability of the data for factor analysis. For this study, values above 0.7 were required for applying EFA (Hair et al., 2014), as the KMO test values varied between 0 and 1. A statistically

significant Bartlett's test ($p < 0.05$) indicated that sufficient correlations existed between the variables to continue with the analysis [35].

## 4. Findings and Discussion

The respondents' age, gender, race/ethnicity, marital status, and education level, as well as the respondents' personal and professional histories, were the descriptive data analysis outcomes. Percentages, averages, and standard deviations were utilised as descriptive statistics.

The characteristics of the 543 respondents are shown in Table 1.

**Table 1.** Profiles of respondents.

| Demographic | Characteristic | Frequency | Percentage |
|---|---|---|---|
| Gender | Male | 472 | 87 |
| | Female | 71 | 13 |
| Age | <20 years | 26 | 4.8 |
| | 20–25 years | 52 | 9.6 |
| | 26–30 years | 148 | 27.3 |
| | 31–35 years | 129 | 23.8 |
| | 36–40 years | 106 | 19.5 |
| | 41–45 years | 73 | 13.4 |
| | 46 years or above | 9 | 1.7 |
| Occupation | Contractors | 120 | 22.1 |
| | Site engineers | 119 | 21.9 |
| | Timekeepers | 120 | 22.1 |
| | Facilitators | 120 | 22.1 |
| | GSOP desk officers | 32 | 5.9 |
| | Director of public works | 32 | 5.9 |
| Education level | Master's degree | 16 | 2.9 |
| | Bachelor's degree | 197 | 36.3 |
| | National diploma | 72 | 13.3 |
| | Technical/SSCE | 166 | 30.6 |
| | Matric certificate/BECE | 92 | 16.9 |
| Experience | 2–5 years | 255 | 47 |
| | 6–10 years | 202 | 37.2 |
| | 11–15 years | 51 | 9.4 |
| | 16–20 years | 22 | 4.1 |
| | 20 years and above | 13 | 2.4 |
| Geographical location | Bolgatanga | 112 | 20.6 |
| | Wa | 112 | 20.6 |
| | Tamale | 110 | 20.3 |
| | Kumasi | 109 | 20.1 |
| | Accra | 100 | 18.4 |

The majority of respondents (87.2%) were male; the median age range was 26–35 (51.1%); and 4.8% of the sample was younger than 20 years old.

Most of the respondents worked as engineers (27.8%), followed by contractors (22.1%), timekeepers (22.1%), and facilitators (22.1%).

The majority of respondents (66.1%) either held a bachelor's degree (36.3%) or a technical certification (30.6%), while 16.9% had completed high school. While nearly half of respondents (47%) had two to five years of experience in the workforce, just over half (53.1%) had six years of experience or more. This demonstrates that the respondents were capable of working in the construction business and had the necessary experience to provide data that may be used to draw conclusions on parameters measuring labour productivity. The geographic distribution of respondents was nearly even: 20.6% were from Bolgatanga, 20.6% were from Wa, 20.3% were from Tamale, 20.1% were from Kumasi, and 18.4% were from Accra. However, the majority of respondents (61.5%) were employed in the three northern regions of Ghana: Bolgatanga, Wa, and Tamale.

### 4.1. Results from Exploratory Factor Analysis

This part of the report details the findings from Questionnaire Section B, which aimed to identify the factors affecting the productivity of construction workers in Ghana's most labour-intensive industry: road building. The results of the means, standard deviations, and the rank of item score of the data, as well as the exploratory factor analysis (EFA) of the results, are presented. The descriptive results revealed the ranking of all the factors from the highest to the lowest and the individual means and standard deviations of the factors.

Table 2 reveals the respondents' rankings of the worker component (WC) that can promote the labour productivity of labour-intensive works on road construction. It shows that "the company's incentive scheme for good performance" was ranked first, with a mean score of 4.12 and a standard deviation (SD) of 0.976; "opportunities for employees to exercise their skills" was ranked second, with a mean score of 4.11 and an SD of 0.697; "likelihood beneficiaries are paid on time" was ranked third, with a mean score of 4.10 and an SD of 0.986; "management response to settle employees' grievances" was ranked fourth, with a mean score of 4.03 and an SD of 1.02; and "beneficiaries' knowledge of scientific techniques" was ranked fifth, with a mean score of 3.95 and an SD of 0.839.

In addition, "beneficiaries' attitude towards the job they have to execute" was ranked sixth, with a mean score of 3.91 and an SD of 0.852; "beneficiaries' knowledge of career prospects" was ranked seventh, with a mean score of 3.88 and an SD of 0.921; "promotion opportunities for employees" was ranked eighth, with a mean score of 3.86 and an SD of 1.16; "employment of young beneficiaries on projects" was ranked ninth, with a mean score of 3.83 and an SD of 1.46; and "Beneficiaries' level of experience to do their work" was ranked tenth, with a mean score of 3.77 and an SD of 0.496.

Furthermore, "likelihood older beneficiaries will be replaced by younger beneficiaries" was ranked eleventh, with a mean score of 3.69 and SD of 1.17; "level of safety achieved on projects" was ranked twelfth, with a mean score of 3.67 and an SD of 0.963; "employment of older beneficiaries from villages" was ranked thirteenth, with a mean score of 3.60 and an SD of 1.23; "employees level of awareness of company policy" was ranked fourteenth, with a mean score of 3.59 and an SD of 0.964; and "incentives used to attract young people into sector" was ranked fifteenth, with a mean score of 3.59 and an SD of 1.73.

Moreover, "the number of multi-skilled project personnel in the company" was ranked sixteenth, with a mean score of 3.57 and an SD of 0.591; "beneficiaries' having formal training in labour-intensive works" was ranked seventeenth, with a mean score of 3.54 and an SD of 0.965; "quality of transportation facilities for beneficiaries" was ranked eighteenth, with a mean score of 3.53 and an SD of 1.057; "the usage of safety wear on-site" was ranked nineteenth, with a mean score of 3.21 and an SD of 0.961; and "degree to which safety standards on a project comply with legislated criteria" was ranked twentieth, with a mean score of 3.08 and an SD of 1.098.

**Table 2.** Descriptive statistics for worker component (WC).

| Factors | Mean | SD | Rank |
|---|---|---|---|
| The company's incentive scheme for good performance | 4.12 | 0.976 | 1 |
| Opportunities for employees to exercise their skills | 4.11 | 0.697 | 2 |
| Likelihood beneficiaries are paid on time | 4.10 | 0.986 | 3 |
| Management response to settle employees' grievances | 4.03 | 1.018 | 4 |
| Beneficiaries' knowledge of scientific techniques | 3.95 | 0.839 | 5 |
| Beneficiaries' attitude towards the job they have to execute | 3.91 | 0.852 | 6 |
| Beneficiaries' knowledge of career prospects | 3.88 | 0.921 | 7 |
| Promotion opportunities for employees | 3.86 | 1.160 | 8 |
| Employment of young beneficiaries on projects | 3.83 | 1.464 | 9 |
| Beneficiaries' level of experience to do their work | 3.77 | 0.496 | 10 |
| Likelihood older beneficiaries will be replaced by younger workers | 3.69 | 1.173 | 11 |
| Level of safety achieved on projects | 3.67 | 0.963 | 12 |
| Employment of older beneficiaries from villages | 3.60 | 1.234 | 13 |
| Employees level of awareness of company policy | 3.59 | 0.964 | 14 |
| Incentives used to attract young people into the sector | 3.59 | 1.727 | 15 |
| The number of multi-skilled project personnel in the company | 3.57 | 0.591 | 16 |
| Beneficiaries' having formal training in labour-intensive works | 3.54 | 0.965 | 17 |
| Quality of transportation facilities for beneficiaries | 3.53 | 1.057 | 18 |
| The usage of safety wear on site | 3.21 | 0.961 | 19 |
| The degree to which safety standards on a project comply with legislated criteria | 3.08 | 1.098 | 20 |

## *4.2. Results from Exploratory Factor Analysis*

The results from the EFA on the worker component that can promote the labour productivity of labour-intensive works for road construction are presented. Of the twenty (20) variables listed, the following five (5) were omitted: "workers' having formal training in labour-intensive works" (WC1), "beneficiaries level of awareness of company policy" (WC6), "beneficiaries' knowledge of career prospects" (WC10), "likelihood beneficiaries are paid on time" (WC12), and "degree to which safety standards on a project comply with legislated criteria"(WC15).

The 20 worker factors that can promote labour productivity of the labour-intensive works for road construction were subjected to PCA to assess their validity and reliability. The results report the suitability of the data to be analysed, the factor extraction and rotation, and the interpretation.

As shown in Table 3, the KMO measure of sampling adequacy achieved a value of 0.892, exceeding the recommended minimum value of 0.7, and Bartlett's test of sphericity was also statistically significant (<0.05), thus supporting the factorability of the data.

**Table 3.** KMO and Bartlett's test for worker component.

| Kaiser–Meyer–Olkin Measure of Sampling Adequacy | | 0.892 |
|---|---|---|
| Bartlett's Test of Sphericity | Approx. Chi-Square | 14,645.327 |
| | df | 190 |
| | Sig. | 0.000 |

The pattern matrix in Table 4 shows that, out of the initial 19 variables, PCA extracted 15 variables in four components with factor loadings above 0.4 with the potential to influence the labour productivity of labour-intensive works for road construction in Ghana.

**Table 4.** Pattern factor loading for worker component.

| Code | Variable | Component | | | |
|------|----------|-----------|---|---|---|
| | | 1 | 2 | 3 | 4 |
| WC19 | Employment of older beneficiaries from villages. | 0.954 | 0.220 | 0.065 | 0.041 |
| WC18 | Employment of young beneficiaries on projects. | 0.914 | −0.279 | 0.001 | 0.126 |
| WC20 | Incentives used to attract young people into sector. | 0.776 | −0.406 | −0.068 | −0.227 |
| WC17 | Likelihood older beneficiaries will be replaced by younger beneficiaries. | 0.774 | −0.134 | 0.300 | 0.260 |
| WC2 | Beneficiaries' level of experience to do their work. | 0.073 | −0.172 | 0.240 | −0.817 |
| WC3 | The number of multi-skilled project personnel in the company. | 0.119 | 0.789 | −0.060 | −0.188 |
| WC4 | Beneficiaries' knowledge of scientific techniques. | −0.081 | 0.717 | 0.307 | −0.330 |
| WC1 | Beneficiaries' having formal training in labour-intensive works. | −0.473 | 0.565 | 0.302 | −0.337 |
| WC6 | Beneficiaries' level of awareness of company policy. | 0.701 | 0.548 | −0.212 | −0.104 |
| WC8 | Opportunities for employees to exercise their skills. | 0.343 | 0.820 | 0.027 | −0.106 |
| WC11 | Beneficiaries' attitude towards the job they have to execute. | 0.034 | 0.927 | 0.055 | 0.142 |
| WC16 | The usage of safety wear on site. | −0.065 | 0.031 | 0.952 | 0.230 |
| WC15 | Degree to which safety standards on a project comply with legislated criteria. | 0.528 | 0.291 | 0.636 | −0.276 |
| WC14 | Level of safety achieved on projects. | −0.253 | 0.198 | 0.798 | −0.165 |
| WC13 | Quality of transportation facilities for workers. | −0.246 | −0.135 | −0.127 | 0.858 |
| WC7 | The company's incentive scheme for good performance. | −0.195 | 0.221 | 0.191 | 0.787 |
| WC10 | Beneficiaries' knowledge of career prospects. | 0.015 | −0.004 | 0.628 | 0.645 |
| WC12 | Likelihood workers are paid on time. | 0.422 | −0.031 | 0.419 | −0.577 |
| WC9 | Management response to settle beneficiaries' grievances. | −0.115 | 0.091 | −0.310 | 0.854 |
| WC5 | Promotion opportunities for beneficiaries. | 0.280 | 0.121 | −0.090 | 0.711 |
| | Extraction method: principal component analysis. | | | | |
| | Rotation method: oblimin with Kaiser normalization. a | | | | |

a: Rotation converged in 11 iterations.

As indicated in Table 5, the eigenvalue was set at a conventional high value of 1.00 [27]. In determining the number of principal components to be extracted, the latent root criterion was applied, which recommends that four components should be extracted because their eigenvalues are greater than one.

**Table 5.** Correlation matrix of factor analysis for worker component.

| | WC1 | WC2 | WC3 | WC4 | WC5 | WC6 | WC7 | WC8 | WC9 | WC10 | WC11 | WC12 | WC13 | WC14 | WC15 | WC16 | WC17 | WC18 | WC19 | WC20 |
|---|---|---|---|---|---|---|---|---|---|---|---|---|---|---|---|---|---|---|---|---|
| WC1 | 1.000 | 0.228 | 0.511 | 0.794 | 0.394 | −0.006 | 0.005 | 0.136 | 0.420 | −0.087 | 0.217 | 0.067 | −0.359 | 0.485 | −0.046 | 0.573 | −0.460 | −0.598 | −0.239 | −0.48 |
| WC2 | 0.228 | 1.000 | 0.258 | 0.352 | 0.566 | 0.315 | 0.418 | 0.452 | 0.540 | 0.342 | 0.223 | 0.500 | 0.377 | 0.474 | 0.18 | −0.047 | 0.195 | 0.126 | 0.237 | 0.287 |
| WC3 | 0.511 | 0.258 | 1.000 | 0.689 | 0.378 | 0.554 | 0.277 | 0.071 | 0.135 | 0.003 | −0.024 | 0.297 | 0.097 | 0.509 | 0.485 | 0.615 | −0.183 | −0.215 | 0.235 | −0.179 |
| WC4 | 0.794 | 0.352 | 0.689 | 1.000 | 0.535 | 0.421 | 0.36 | 0.311 | 0.524 | 0.158 | 0.201 | 0.298 | −0.02 | 0.543 | 0.134 | 0.589 | −0.167 | −0.347 | 0.135 | −0.227 |
| WC5 | 0.394 | 0.566 | 0.378 | 0.535 | 1.000 | 0.537 | 0.652 | 0.232 | 0.193 | 0.23 | −0.116 | 0.401 | 0.467 | 0.463 | 0.425 | 0.065 | 0.125 | 0.197 | 0.420 | 0.326 |
| WC6 | −0.006 | 0.315 | 0.554 | 0.421 | 0.537 | 1.000 | 0.709 | 0.169 | 0.035 | 0.295 | −0.177 | 0.36 | 0.598 | 0.273 | 0.65 | 0.354 | 0.316 | 0.370 | 0.700 | 0.317 |
| WC7 | 0.005 | 0.418 | 0.277 | 0.360 | 0.652 | 0.709 | 1.000 | 0.569 | 0.305 | 0.705 | 0.219 | 0.591 | 0.701 | 0.100 | 0.397 | 0.009 | 0.564 | 0.661 | 0.813 | 0.626 |
| WC8 | 0.136 | 0.452 | 0.071 | 0.311 | 0.232 | 0.169 | 0.569 | 1.000 | 0.737 | 0.839 | 0.762 | 0.673 | 0.359 | −0.034 | −0.207 | −0.126 | 0.496 | 0.367 | 0.466 | 0.302 |
| WC9 | 0.420 | 0.540 | 0.135 | 0.524 | 0.193 | 0.035 | 0.305 | 0.737 | 1.000 | 0.551 | 0.715 | 0.501 | −0.025 | 0.268 | −0.4 | 0.077 | 0.204 | −0.072 | 0.115 | −0.008 |
| WC10 | −0.087 | 0.342 | 0.003 | 0.158 | 0.230 | 0.295 | 0.705 | 0.839 | 0.551 | 1.000 | 0.624 | 0.677 | 0.592 | −0.143 | 0.015 | −0.185 | 0.670 | 0.628 | 0.688 | 0.578 |
| WC11 | 0.217 | 0.223 | −0.024 | 0.201 | −0.116 | −0.177 | 0.219 | 0.762 | 0.715 | 0.624 | 1.000 | 0.461 | −0.032 | −0.18 | −0.457 | −0.038 | 0.277 | 0.123 | 0.159 | 0.065 |
| WC12 | 0.067 | 0.500 | 0.297 | 0.298 | 0.401 | 0.360 | 0.591 | 0.673 | 0.501 | 0.677 | 0.461 | 1.000 | 0.58 | 0.241 | 0.195 | −0.165 | 0.350 | 0.408 | 0.605 | 0.507 |
| WC13 | −0.359 | 0.377 | 0.097 | −0.02 | 0.467 | 0.598 | 0.701 | 0.359 | −0.025 | 0.592 | −0.032 | 0.58 | 1.000 | 0.008 | 0.559 | −0.246 | 0.589 | 0.777 | 0.809 | 0.809 |
| WC14 | 0.485 | 0.474 | 0.509 | 0.543 | 0.463 | 0.273 | 0.100 | −0.034 | 0.268 | −0.143 | −0.18 | 0.241 | 0.008 | 1.000 | 0.384 | 0.360 | −0.413 | −0.331 | −0.002 | −0.046 |
| WC15 | −0.046 | 0.180 | 0.485 | 0.134 | 0.425 | 0.650 | 0.397 | −0.207 | −0.400 | 0.015 | −0.457 | 0.195 | 0.559 | 0.384 | 1.000 | 0.249 | −0.020 | 0.300 | 0.487 | 0.388 |
| WC16 | 0.573 | −0.047 | 0.615 | 0.589 | 0.065 | 0.354 | 0.009 | −0.126 | 0.077 | −0.185 | −0.038 | −0.165 | −0.246 | 0.360 | 0.249 | 1.00 | −0.264 | −0.398 | 0.03 | −0.424 |
| WC17 | −0.460 | 0.195 | −0.183 | −0.167 | 0.125 | 0.316 | 0.564 | 0.496 | 0.204 | 0.67 | 0.277 | 0.35 | 0.589 | −0.413 | −0.020 | −0.264 | 1.000 | 0.789 | 0.665 | 0.640 |
| WC18 | −0.598 | 0.126 | −0.215 | −0.347 | 0.197 | 0.370 | 0.661 | 0.367 | −0.072 | 0.628 | 0.123 | 0.408 | 0.777 | −0.331 | 0.300 | −0.398 | 0.789 | 1.000 | 0.802 | 0.874 |
| WC19 | −0.239 | 0.237 | 0.235 | 0.135 | 0.42 | 0.700 | 0.813 | 0.466 | 0.115 | 0.688 | 0.159 | 0.605 | 0.809 | −0.002 | 0.487 | 0.030 | 0.665 | 0.802 | 1.000 | 0.731 |
| WC20 | −0.480 | 0.287 | −0.179 | −0.227 | 0.326 | 0.317 | 0.626 | 0.302 | −0.008 | 0.578 | 0.065 | 0.507 | 0.809 | −0.046 | 0.388 | −0.424 | 0.640 | 0.874 | 0.731 | 1.000 |

In Table 6, a Cronbach alpha of 0.876 was obtained for the worker component. This satisfied the benchmark provided in [26,35], which states that the Cronbach alpha coefficient of a scale should be above 0.7. The closer the alpha ($\alpha$) is to 1, the greater the internal consistency of items in the instrument is assumed to be.

**Table 6.** Cronbach's alpha co-efficient for worker component.

| Reliability Statistics | | |
|---|---|---|
| Cronbach's Alpha | Cronbach's Alpha Based on Standardized Items | N of Items |
| 0.876 | 0.884 | 20 |

Table 7 shows that after rotation, four components with eigenvalues exceeding 1.0 were extracted and were meaningful to retain. Factor one explains 36.43% of the total variance; factor two, 23.29%; factor three, 16.81%; and factor four, 6.79%. Thus, the final statistics of the PCA shows that three extracted factors explain a cumulative variance of approximately 83.32%.

**Table 7.** Total variance explained for worker component.

| Component | Initial Eigenvalues | | | Extraction Sums of Squared Loadings | | | Rotation Sums of Squared Loadings |
|---|---|---|---|---|---|---|---|
| | Total | % of Variance | Cumulative % | Total | % of Variance | Cumulative % | Total |
| 1 | 7.286 | 36.430 | 36.430 | 7.286 | 36.430 | 36.430 | 6.703 |
| 2 | 4.659 | 23.294 | 59.723 | 4.659 | 23.294 | 59.723 | 3.932 |
| 3 | 3.362 | 16.811 | 76.534 | 3.362 | 16.811 | 76.534 | 3.982 |
| 4 | 1.358 | 6.789 | 83.323 | 1.358 | 6.789 | 83.323 | 4.037 |
| 5 | 0.726 | 3.629 | 86.952 | | | | |
| 6 | 0.560 | 2.802 | 89.754 | | | | |
| 7 | 0.420 | 2.098 | 91.852 | | | | |
| 8 | 0.342 | 1.712 | 93.564 | | | | |
| 9 | 0.260 | 1.301 | 94.865 | | | | |
| 10 | 0.207 | 1.037 | 95.902 | | | | |
| 11 | 0.175 | 0.874 | 96.776 | | | | |
| 12 | 0.146 | 0.731 | 97.507 | | | | |
| 13 | 0.110 | 0.551 | 98.058 | | | | |
| 14 | 0.082 | 0.410 | 98.468 | | | | |
| 15 | 0.081 | 0.404 | 98.873 | | | | |
| 16 | 0.064 | 0.321 | 99.194 | | | | |
| 17 | 0.057 | 0.287 | 99.481 | | | | |
| 18 | 0.042 | 0.208 | 99.689 | | | | |
| 19 | 0.039 | 0.197 | 99.886 | | | | |
| 20 | 0.023 | 0.114 | 100.000 | | | | |

Extraction method: principal component analysis.

Table 8 reveals the correlation of variables based on their factor loadings after rotation in PCA. Three components with eigenvalues above 1, as shown in Table 4, were examined for the inherent relationships among the variables under each factor. Variables with the highest factor loading in one component belonged to that component; the highest factor loading had to be of a significant value of 0.4 or above (see Table 4). Component 1 was labelled age of a beneficiary; Component 2 was labelled beneficiaries' knowledge; Component 3 was labelled safety compliance; and Component 4 was labelled motivation of beneficiaries. The names given to these factors were derived from a close examination of the variables within each of the factors.

**Table 8.** Rotated factor matrix [a] for worker component.

| Code | Variable | Component | | | |
|---|---|---|---|---|---|
| | | **1** | **2** | **3** | **4** |
| WC 19 | Employment of older beneficiaries from villages. | 0.954 | | | |
| WC18 | Employment of young beneficiaries on projects. | 0.914 | | | |
| WC17 | Likelihood older beneficiaries will be replaced by younger beneficiaries. | 0.774 | | | |
| WC20 | Incentives used to attract young people into the sector. | 0.776 | | | |
| WC6 | Beneficiaries' level of awareness of company policy. | | 0.701 | | |
| WC3 | The number of multi-skilled project personnel in the company. | | 0.789 | | |
| WC4 | Beneficiaries' knowledge of scientific techniques. | | 0.717 | | |
| WC1 | Beneficiaries' having formal training in labour-intensive works. | | 0.565 | | |
| WC11 | Beneficiaries' attitude towards the job they have to execute. | | 0.927 | | |
| WC8 | Opportunities for beneficiaries to exercise their skills. | | 0.820 | | |
| WC2 | Beneficiaries' level of experience to do their work. | | 0.817 | | |
| WC16 | The usage of safety wear on site. | | | 0.952 | |
| WC15 | Degree to which safety standards on a project comply with legislated criteria. | | | 0.636 | |
| WC14 | Level of safety achieved on projects. | | | 0.798 | |
| WC13 | Quality of transportation facilities for beneficiaries. | | | | 0.858 |
| WC9 | Management response to settle employees' grievances. | | | | 0.854 |
| WC7 | The company's incentive scheme for good performance. | | | | 0.787 |
| WC5 | Promotion opportunities for employees. | | | | 0.711 |
| WC10 | Beneficiaries' knowledge of career prospects. | | | | 0.645 |
| WC12 | Likelihood workers are paid on time. | | | | −0.577 |
| | Extraction method: principal component analysis. | | | | |
| | Rotation method: oblimin with Kaiser normalization. a | | | | |

a: Rotation converged in 11 iterations.

The scree plot presented in Figure 1 also reveals the excluded factors by indicating the cut-off point at which the eigenvalues levelled off.

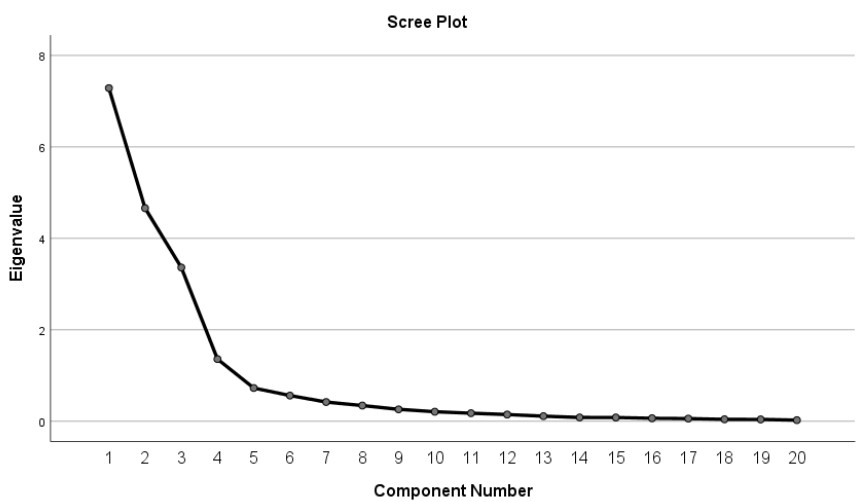

**Figure 1.** Scree plot for factor analysis.

In Table 7, four (4) components with eigenvalues greater than one were found using principal axis factoring. The following interpretations were produced in light of the evaluation of the underlying correlations between the variables under each factor. The age of beneficiaries, beneficiaries' knowledge, safety compliance, and motivation of beneficiaries were the four categories under which Component 1 was classified. After carefully analysing the variables contained inside each of the elements, these component names were developed.

Below is an explanation of the individual indications that made up each of the three extracted components, along with a description of how each was characterized in detail during focus group sessions.

**Component 1: Age of Beneficiaries**

Table 7 shows the four (4) WC variables that were retrieved from the workers. The Component 1 measures were "employment of older beneficiaries from villages" (95.4%), "employment of young beneficiaries on projects" (91.4%), "incentives utilized to lure young people into sector" (77.6%), and "likelihood older beneficiaries will be replaced by younger people" (76.6%). The numbers in brackets represent the loadings on the corresponding factors. Approximately 36.4 percent of the overall variation may be attributed to this group.

**Component 2: Beneficiaries' Knowledge**

The seven variables that were extracted from the worker component (WC) for Component 2 were as follows: "beneficiaries' attitude towards the job they have to execute" (92.7%), "opportunities for employees to exercise their skills" (82%), "employees' level of experience to do their work" (81.7%), "the number of multi-skilled project personnel in the company" (78.9%), "beneficiaries' knowledge of scientific techniques" (71.7%), "Beneficiaries' level of awareness of company policy" (70.1%) and "Beneficiaries' having formal training in labour-intensive works" (56.5%). The numbers in parentheses denote the relevant factor loadings for that factor. This cluster was responsible for 23.3% of the total variation in the data. The results agree with the findings in [41–47], which indicated that the knowledge of the workers on the task is very important when it comes to achieving productivity for road construction.

**Component 3: Safety Compliance**

This group explained 16.8% of the total variation. The Component 3 variables that were retrieved from workers were as follows (Table 7): "use of safety wear on-site" (95.2%), "level of safety reached on projects" (79.8%), and "degree to which safety requirements on a project meet with legislated criteria" (63.6%). The numbers in parentheses represent factor loadings.

**Component 4: Motivation of Beneficiaries**

The factors "quality of transportation facilities for workers" (85.8%), "management response to settle employees' grievances" (85.4%), "the company's incentive scheme for good performance" (78.7%), "promotion opportunities for employees" (71.71), "beneficiaries' knowledge of career prospects" (64.5%), and "likelihood beneficiaries are paid on time" (44.7%) were the six (6) extracted worker component (WC) variables for Component 4, as shown in Table 7. The numbers in parentheses represent the factor loadings. This cluster was responsible for 6.8% of the variation.

These results confirm the findings of the authors of [48,49], who stressed that there is a need for the workers to comply with construction specifications, ensuring the quality of the work delivered. Owing to the industry's labour-intensive nature, the workforce factor plays a significant role in the construction project implementation process [50]. The authors of [51–55] confirmed that workforce accounts for 30–50% of the total project cost. Consequently, considering the worker component's key role in achieving a higher level of productivity performance, construction professionals should pay more attention to the workforce dimension, which has four observable variables in line with the relevant

findings: (i) age of beneficiaries [56–59]; (ii) beneficiaries' knowledge [60,61]; (iii) safety compliance [54]; and (iv) motivation of beneficiaries [48].

This research is novel because it establishes a framework for measuring the productivity of workers in the labour-intensive sector of the Ghanaian construction industry by looking at variables such as the age of beneficiaries, the knowledge of beneficiaries, compliance with safety regulations, and the motivation of beneficiaries as illustrated in Figure 2. A similar approach, based on the latent factors that were used to derive the labour productivity result variables, might be utilised to assess company-level productivity in the Ghanaian construction industry.

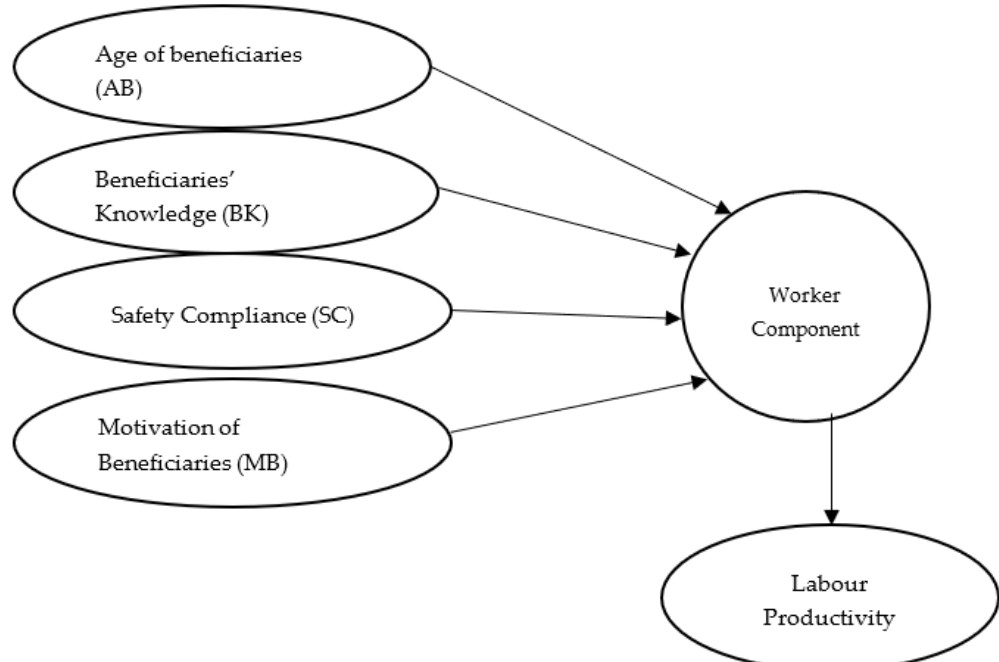

**Figure 2.** Framework for productivity of construction workers on labour-intensive works.

Construction industry experts might also utilise this information to better guide company-wide strategies. The study's findings can also be used as a useful tool in planning to expedite the efficient utilisation of labour-intensive road building activity to boost productivity and meet contractual obligations. It can also help contractors estimate how long it will take to construct a certain road using the labour-intensive way.

This research has shown that the four components have a significant impact on the labour productivity of indigenous construction enterprises operating in Ghana. The findings could be used by the Association of Building and Civil Engineering Contractors of Ghana (ABCECG) to prioritise criteria for boosting the efficiency of local construction companies in Ghana. In the end, this will help the ABCECG determine where they need to focus their efforts in order to build up the capacity of their contractors.

As a reference, the framework can be used to make sure that road construction companies have all they need to have a highly productive workforce.

## 5. Conclusions

Road construction is a labour-intensive industry; hence, a framework for assessing labour productivity in this sector was established using existing theories from the field of study. The theory postulates that exogenous (latent) factors have an impact on how well construction businesses' total workforces perform in terms of their labour productivity.

SPSS version 24.0 (SPSS version 24.0 Inc., Chicago, IL, USA) for Windows and Microsoft Excel were used to construct frequencies, tables, charts, figures, and cross-tabulations for the proposed framework analysis. The framework diagram was created using Amos ver-

sion 23 (AMOS Group Limited, Singapore). The measurement and framework fit statistics were in good agreement with the sample data.

It was found that worker components significantly influenced the output of Ghanaian construction workers in labour-intensive road building projects. This verdict was based on the conclusive empirical approach that was developed. Therefore, the four-factor paradigm adequately characterised the labour productivity of Ghana's construction sector for tasks that rely heavily on human effort, such as road construction.

Both the theoretical and practical ramifications of the current study's findings are discussed to demonstrate the study's significance and contribution.

This study fills a vacuum in our understanding of which factors are most important for predicting productivity in Ghana's labour-intensive road construction industry, and this alone is a significant contribution. The results show that the labour productivity of businesses is multifaceted and intricate. The results of the exploratory factor analysis show that the latent factors led to labour productivity outcome variables, which may be used to measure the productivity of a company's workforce.

The influence of the workforce on labour-intensive road building jobs is discussed for construction professionals who want to give their companies a competitive edge in the construction sector through increased productivity.

The age of beneficiaries, understanding of beneficiaries, compliance with safety regulations, and motivation of beneficiaries all play key roles. Construction personnel can also use this data to aid in making business decisions. This study is also helpful as a planning tool for implementing labour-intensive techniques that boost productivity and enable the meeting of contract deadlines through the on-time completion of an assignment.

It is hoped that this research will aid policymakers in the construction industry as they update the country's labour-intensive public works strategy to better support indigenous firms and boost construction productivity.

To take into consideration the many people who would profit from labour-intensive implementation, a method statement must be employed. It has been discovered at several sites that increasing the number of project beneficiaries may decrease the overall output (due to the overcrowding of workers). Government officials hope to use this platform to reach out to the area's poor, so potential new members should tread carefully.

## 6. Limitations of the Study

Although interesting and valuable findings have emerged from this study, it is not without limitations. Therefore, the following limitations regarding this current study were experienced. Firstly, the study was focused only on road construction where labour-intensive works took place during the period under consideration and covered 40 districts in Ghana. Given enough resources, it would be preferable to conduct a similar study in the whole Ghanaian territory in other aspects of labour-intensive works, such as dams, dugouts, or building construction.

Secondly, the EFA analysis used to analyse the generated data was construed as a limitation. The results presented herein are based on the analysis of a framework with the new measurement. Thirdly, although the internal reliability tests indicated high internal consistency, and therefore a well-constructed research tool, some constructs revealed high correlational values. This may be because only one questionnaire was used to collect information from the research respondents.

A final limitation is related to the sample size. All empirical studies are limited by the nature of the sample studied. The exploration of the dependent variable (firms' labour productivity) has shown that it is multi-faceted, and claims further interpretations.

**Author Contributions:** Conceptualization, E.B.-A., D.W.T. and C.A.; methodology, E.B.-A., D.W.T. and C.A.; software, E.B.-A., D.W.T. and C.A.; validation, E.B.-A., D.W.T. and C.A.; formal analysis, E.B.-A., D.W.T. and C.A.; investigation, E.B.-A., D.W.T. and C.A.; resources, E.B.-A., D.W.T. and C.A.; data curation, E.B.-A., D.W.T. and C.A.; writing—original draft preparation, E.B.-A., D.W.T. and C.A.; writing—review and editing, E.B.-A., D.W.T. and C.A.; visualization, E.B.-A., D.W.T. and C.A.; supervision, E.B.-A., D.W.T. and C.A. All authors have read and agreed to the published version of the manuscript.

**Funding:** This research received no external funding.

**Data Availability Statement:** Not applicable.

**Conflicts of Interest:** The authors declare no conflict of interest.

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
