# Peer review of "Performance Improvement of Construction Workers to Achieve Better Productivity for Labour-Intensive Works"

_buildings, doi:10.3390/buildings12101593_

Round 1

Reviewer 1 Report

Read my comments in the attached file

Author Response

I have worked on the methodology and the result sections as suggested. Attached the response.

Reviewer 2 Report

The productivity-related theories presented in previous studies are well organized, but the differences between the implications of this study and existing studies are not clear. In addition, the system of manuscript is generally insufficient.

If the abbreviation of Factor is displayed together in Figure 1, it seems to be helpful in understanding the contents.

It is necessary to reconsider whether the distinction between 2.2, 2.3, and 2.4 is absolutely necessary. The contents of each section are quite short, so it is not appropriate to separate them into separate sections.

The measurements in Table 1 do not include all the measurements presented in 2.3. What process was used to derive Table 1?

What does '71 Likert scale items' mean? do you mean scale? Do you mean the number of measurements? If it means a scale, it is correct to express it on 5 Likert scale or 7 Likert scale.

Are the respondents' statistics presented between 4 and 4.1 significant enough to occupy a page? It is judged appropriate to create '4.1 Respondent Statistics' and express the statistics in a simple table.

Are 4.2~4.5 results for descriptive statistics and 4.6~4.9 results from factor analysis? Rather than listing them at the same level, it would be better to organize 4.2-4.5 as one group and 4.6-4.9 as one group.

Is Table 2 the result derived from the previous descriptive statistical analysis and factor analysis results? Please present it in connection with the analysis results. And is there a reason why it says 'Climatic' here rather than 'Temperature'?

The content of 5.1 is too poor.

Author Response

Thanks, I have responded to the comments. I have attached the response. Thank you.

Reviewer 3 Report

REVIEW

on article

Productivity of Construction Workers Using Labour Intensive Works

Bamfo-Agyei, Emmanuel, Thwala Didibhuku Wellington and Aigbavboa, Clinton.

SUMMARY

The article submitted for review examines the performance of builders using labor-intensive work. The article is interesting; it develops methods for increasing labor productivity in the construction industry and determining the level of labor productivity in the Ghanaian construction industry. Thus, the study solves an technical problem and is aimed at a specific goal and objectives.

The authors applied interesting methods in the study, a quantitative design of the study was adopted, in which a questionnaire was used. The authors applied a stratified sampling technique that is based on the diverse nature of the population. The authors conducted a rather large study, analyzed the data obtained, and as a result of this, they obtained specific conclusions.

However, in addition to the advantages of the article, there are several serious shortcomings that should be corrected. After they are corrected, the article will need to be sent for re-review, because in its current form it cannot yet be published in Buildings journal. Below are the comments of the reviewer.

COMMENTS

1.    The title of the article is too general and does not reflect the specifics of the study that the authors were involved in. The title of the article should be revised to reflect the peculiarity of the study and emphasize its main result. Perhaps the title should be given some terms like performance improvement or performance improvement.

2.    The summary is presented vaguely and unclearly. There is no specific research problem, it is not clear why the research was conducted.

3.    A lot of text in the abstract describes the methods. More specific information about the methods should be given, while paying attention to the scientific novelty, the scientific result of the study.

4.    The abstract contains phrases that speak only about the enumeration of the operations performed, but do not contain the specific result of the work done by the authors. That is, the abstract in its current form does not reflect the content of the article, the abstract should be seriously revised.

5.    The Introduction is very concise. The authors analyzed only 7 sources in the Introduction. This is very little to talk about such a problem as increasing labor productivity in construction. The Introduction should be seriously revised, increasing the number of analyzed sources to at least 15-20 titles. At the same time, the Introduction should end with a clearer formulation of the purpose and objectives of the study, as well as the problem that follows from the review.

6.    Very large section 2 suggests that the authors have heavily theorized this article. Secondly, such section 2 looks somewhat ponderous. Probably, it was necessary to give a block diagram of the conducted research with a clear program of theoretical and practical research before section 2.

7.    In addition, section 2 should have a smoother transition to the Materials and Methods section. That is, there should be a clear relationship between the theoretical and practical parts of the article.

8.    If the authors declare in section 3 the heading "Materials and Methods", then this paragraph should be clearly divided into 2 parts: "Materials" and separately "Methods". The authors have divided this section into 6 small subsections, and it looks rather heavy, so this section should also be structured. Now, this can be perceived by readers very hard.

9.    In section 4, attention is drawn to numerous graphs of the same type with minimal explanations, we are talking about figures 3, 4, 5, 6, 8. It should be explained in more detail what the authors want to bring with such a large amount of graphic material and, most importantly, how they interpret it and what intermediate conclusions follow from these drawings.

10.  Table 1 in subsection 4.1 looks cumbersome, but not very informative. Perhaps it should be presented in some more systematic form. It considers the ranking of factors of various components that affect labor productivity. Maybe it makes sense to restructure the tabular view into another analytical view with a more visual form, for example, in the form of an Ishikawa cause-and-effect diagram.

11.  The same remark can be applied to tables 2 and 3. However, this is at the discretion of the authors.

12.  The authors did not provide a detailed comparison of their results with those of other authors and did not provide a Discussion section. This is unacceptable, because without the Discussion section and a qualitative comparison of the results with those obtained earlier, it is impossible to establish the scientific novelty and scientific result of the authors.

13.  The conclusions are also divided by the authors into subsections, which somewhat complicates their perception.

14.  The general comment of the reviewer on the article is as follows: the authors touched on an interesting topic, but the presentation of the results should be seriously reworked. The article needs to be reviewed, corrected all the comments made and sent for re-reviewing. In its current form, the article cannot be published in the journal, since the scientific result, practical significance, and compliance of the subject of the article with the profile of the journal are not clear enough.

Author Response

Comments have been addressed. Please see the attachment for the response.

Round 2

Reviewer 2 Report

Please check the review comments.

Author Response

I have addressed the suggestions of the reviewer.

Reviewer 3 Report

1. I recommend the authors redo the Introduction, removing the early theories of Taylor, Marx, and Smith, since you do not refer to them further. Or in the Discussion, compare your results with the results of these theories.

2. The article needs to strengthen the theoretical part. Give the mathematical statement of the problem. If you use factor analysis, do it fully with all statistical criteria, significance assessment, statistically valid conclusions.

3. Very poor presentation of results.

Author Response

I have addressed the comment of the reviewer

Round 3

Reviewer 2 Report

The answers to both the first review and the second review were not enough.

It is difficult to confirm implications from descriptive statistics and factor analysis results. It seems necessary to present richer implications by proceeding to empirical verification of the 'Framework' presented after the factor analysis in the manuscript.

Author Response

The implications have been improved and linked to the framework.

Reviewer 3 Report

Minor corrections regarding the formatting of Tables and References are needed.

Author Response

Tables 4 and 7 are formatted as suggested and references improved.
